# A Near-Linear Time Algorithm for the Chamfer Distance

**Ainesh Bakshi**
MIT
ainesh@mit.edu

**Piotr Indyk**
MIT
indyk@mit.edu

**Rajesh Jayaram**
Google Research
rkjayaram@google.com

**Sandeep Silwal**
MIT
silwal@mit.edu

**Erik Waingarten**
University of Pennsylvania
ewaingar@seas.upenn.edu

## Abstract

For any two point sets $A, B \subset \mathbb{R}^d$ of size up to $n$, the Chamfer distance from $A$ to $B$ is defined as $\mathrm{CH}(A, B) = \sum_{a \in A} \min_{b \in B} d_X(a, b)$, where $d_X$ is the underlying distance measure (e.g., the Euclidean or Manhattan distance). The Chamfer distance is a popular measure of dissimilarity between point clouds, used in many machine learning, computer vision, and graphics applications, and admits a straightforward $\mathcal{O}(dn^2)$-time brute force algorithm. Further, the Chamfer distance is often used as a proxy for the more computationally demanding Earth-Mover (Optimal Transport) Distance. However, the *quadratic* dependence on $n$ in the running time makes the naive approach intractable for large datasets.

We overcome this bottleneck and present the first $(1+\varepsilon)$-approximate algorithm for estimating the Chamfer distance with a near-linear running time. Specifically, our algorithm runs in time $\mathcal{O}(nd \log(n)/\varepsilon^2)$ and is implementable. Our experiments demonstrate that it is both accurate and fast on large high-dimensional datasets. We believe that our algorithm will open new avenues for analyzing large high-dimensional point clouds. We also give evidence that if the goal is to *report* a $(1 + \varepsilon)$-approximate mapping from $A$ to $B$ (as opposed to just its value), then any sub-quadratic time algorithm is unlikely to exist.

## 1 Introduction

For any two point sets $A, B \subset \mathbb{R}^d$ of sizes up to $n$, the Chamfer distance[1] from $A$ to $B$ is defined as

$$\mathrm{CH}(A, B) = \sum_{a \in A} \min_{b \in B} d_X(a, b)$$

where $d_X$ is the underlying distance measure, such as the Euclidean or Manhattan distance. The Chamfer distance, and its weighted generalization called Relaxed Earth Mover Distance [16, 7], are popular measures of dissimilarity between point clouds. They are widely used in machine learning (e.g., [16, 23]), computer vision (e.g., [8, 22, 12, 15]) and computer graphics [17]. Subroutines for computing Chamfer distances are available in popular libraries, such as Tensorflow [3], Pytorch [2] and PDAL [1]. In many of those applications (e.g., [16]) Chamfer distance is used as a faster proxy for the more computationally demanding Earth-Mover (Optimal Transport) Distance.

---

[1]This is the definition adopted, e.g., in [8]. Some other papers, e.g., [12], replace each distance term $d_X(a, b)$ with its square, e.g., instead of $\|a - b\|_2$ they use $\|a - b\|_2^2$. In this paper we focus on the first definition, as it emphasizes the connection to Earth Mover Distance and its relaxed weighted version in [16, 7].

37th Conference on Neural Information Processing Systems (NeurIPS 2023).

Despite the popularity of Chamfer distance, the naïve algorithm for computing it has quadratic $\mathcal{O}(n^2)$ running time, which makes it difficult to use for large datasets. Faster approximate algorithms can be obtained by performing $n$ exact or approximate nearest neighbor queries, one for each point in $A$. By utilizing the state of the art approximate nearest neighbor algorithms, this leads to $(1 + \varepsilon)$-approximate estimators with running times of $\mathcal{O}\big(n(1/\varepsilon)^{\mathcal{O}(d)} \log n\big)$ in low dimensions [6] or roughly $\mathcal{O}\Big(dn^{1 + \frac{1}{2(1+\varepsilon)^2 - 1}}\Big)$ in high dimensions [5]. Alas, the first bound suffers from exponential dependence on the dimension, while the second bound is significantly subquadratic only for relatively large approximation factors.

## 1.1 Our Results

In this paper we overcome this bottleneck and present the first $(1 + \varepsilon)$-approximate algorithm for estimating Chamfer distance that has a *near-linear* running time, both in theory and in practice. Concretely, our contributions are as follows:

- When the underlying metric $d_X$ is defined by the $\ell_1$ or $\ell_2$ norm, we give an algorithm that runs in time $\mathcal{O}\big(nd \log(n)/\varepsilon^2\big)$ and estimates the Chamfer distance up to $1 \pm \varepsilon$ with $99\%$ probability (see Theorem 2.1). In general, our algorithm works for any metric $d_X$ supported by Locality-Sensitive Hash functions (see Definition 2.2), with the algorithm running time depending on the parameters of those functions. Importantly, the algorithm is quite easy to implement, see Figures 1 and 2.

- For the more general problem of *reporting* a mapping $g : A \to B$ whose cost $\sum_{a \in A} d_X(a, g(a))$ is within a factor of $1 + \varepsilon$ from $\mathrm{CH}(A, B)$, we show that, under a popular complexity-theoretic conjecture, an algorithm with a running time analogous to that of our *estimation* algorithm does not exist, even when $d_X(a, b) = \|a - b\|_1$. Specifically, under a Hitting Set Conjecture [24], any such algorithm must run in time $\Omega(n^{2-\delta})$ for any constant $\delta > 0$, even when the dimension $d = \Theta(\log^2 n)$ and $\varepsilon = \frac{\Theta(1)}{d}$. (In contrast, our estimation algorithm runs in near-linear time for such parameters). This demonstrates that, for the Chamfer distance, estimation is significantly easier than reporting.

- We experimentally evaluate our algorithm on real and synthetic data sets. Our experiments demonstrate the effectiveness of our algorithm for both low and high dimensional datasets and across different dataset scales. Overall, it is much faster (**>5x**) than brute force (even accelerated with KD-trees) and both faster and more sample efficient (**5-10x**) than simple uniform sampling. We demonstrate the scalability of our method by running it on *billion-scale* Big-ANN-Benchmarks datasets [21], where it runs up to **50x** faster than optimized brute force. In addition, our method is robust to different datasets: while uniform sampling performs reasonably well for some datasets in our experiments, it performs poorly on datasets where the distances from points in $A$ to their neighbors in $B$ vary significantly. In such cases, our algorithm is able to adapt its importance sampling probabilities appropriately and obtain significant improvements over uniform sampling.

## 2 Algorithm and Analysis

In this section, we establish our main result for estimating Chamfer distance:

**Theorem 2.1** (Estimating Chamfer Distance in Nearly Linear Time). *Given as input two datasets $A, B \subset \mathbb{R}^d$ such that $|A|, |B| \leqslant n$, and an accuracy parameter $0 < \varepsilon < 1$, $\texttt{Chamfer-Estimate}$ runs in time $\mathcal{O}\big(nd \log(n)/\varepsilon^2\big)$ and outputs an estimator $\eta$ such that with probability at least $99/100$,*

$$(1 - \varepsilon)\mathrm{CH}(A, B) \leqslant \eta \leqslant (1 + \varepsilon)\mathrm{CH}(A, B),$$

*when the underlying metric is Euclidean $(\ell_2)$ or Manhattan $(\ell_1)$ distance.*

For ease of exposition, we make the simplifying assumption that the underlying metric is Manhattan distance, i.e. $d_X(a, b) = \|a - b\|_1$. Our algorithm still succeeds whenever the underlying metric admits a locality-sensitive hash function (the corresponding analogue of Definition 2.2).

**Definition 2.2** (Hashing at every scale). There exists a fixed constant $c_1 > 0$ and a parameterized family $\mathcal{H}(r)$ of functions from $X$ to some universe $U$ such that for all $r > 0$, and for every $x, y \in X$

    1. Close points collide frequently:

$$\Pr_{\boldsymbol{h} \sim \mathcal{H}(r)} [\boldsymbol{h}(x) \neq \boldsymbol{h}(y)] \leqslant \frac{\|x - y\|_1}{r},$$

Subroutine Chamfer-Estimate$(A, B, T)$
**Input:** Two subsets $A, B \subset \mathbb{R}^d$ of size at most $n$, and a parameter $T \in \mathbb{N}$.
**Output:** A number $\boldsymbol{\eta} \in \mathbb{R}_{\geqslant 0}$.
1. Execute the algorithm CrudeNN$(A, B)$, and let the output be a set of positive real numbers $\{\boldsymbol{D}_a\}_{a \in A}$ which always satisfy $\boldsymbol{D}_a \geqslant \min_{b \in B} \|a - b\|_1$. Let $\boldsymbol{D} := \sum_{a \in A} \boldsymbol{D}_a$.
2. Construct the probability distribution $\mathcal{D}$, supported on the set $A$, which satisfies that for every $a \in A$,

$$\Pr_{\boldsymbol{x} \sim \mathcal{D}} [\boldsymbol{x} = a] := \frac{\boldsymbol{D}_a}{\boldsymbol{D}}.$$

3. For $\ell \in [T]$, sample $\boldsymbol{x}_\ell \sim \mathcal{D}$ and spend $\mathcal{O}(|B|d)$ time to compute

$$\boldsymbol{\eta}_\ell := \frac{\boldsymbol{D}}{\boldsymbol{D}_{\boldsymbol{x}_\ell}} \cdot \min_{b \in B} \|\boldsymbol{x}_\ell - b\|_1.$$

4. Output

$$\boldsymbol{\eta} := \frac{1}{T} \sum_{\ell=1}^{T} \boldsymbol{\eta}_\ell.$$

Figure 1: The Chamfer-Estimate Algorithm.

2. Far points collide infrequently:

$$\Pr_{\boldsymbol{h} \sim \mathcal{H}(r)} [\boldsymbol{h}(x) = \boldsymbol{h}(y)] \leqslant \exp\left(-c_1 \cdot \frac{\|x - y\|_1}{r}\right).$$

**Uniform vs Importance Sampling.** A natural algorithm for estimating $\mathrm{CH}(A, B)$ proceeds by *uniform sampling*: sample an $a \in A$ uniformly at random and explicitly compute $\min_{b \in B} \|a - b\|_1$. In general, we can compute the estimator $\hat{z}$ for $\mathrm{CH}(A, B)$ by averaging over $s$ uniformly chosen samples, resulting in runtime $\mathcal{O}(nds)$. It is easy to see that the resulting estimator is un-biased, i.e. $\mathbf{E}[\hat{z}] = \mathrm{CH}(A, B)$. However, if a small constant fraction of elements in $A$ contribute significantly to $\mathrm{CH}(A, B)$, then $s = \Omega(n)$ samples could be necessary to obtain, say, a 1% relative error estimate with constant probability. Since each sample requires a linear scan to find the nearest neighbor, this would result in a quadratic runtime.

While such an approach has good empirical performance for well-behaved datasets, it does not work for data sets where the distribution of the distances from points in $A$ to their nearest neighbors in $B$ is skewed. Further, it is computationally prohibitive to verify the quality of the approximation given by uniform sampling. Towards proving Theorem 2.1, it is paramount to obtain an algorithm that works regardless of the structure of the input dataset.

A more nuanced approach is to perform *importance sampling* where we sample $a \in A$ with probability proportional to its contribution to $\mathrm{CH}(A, B)$. In particular, if we had access to a distribution, $\boldsymbol{D}_a$, over elements $a \in A$ such that, $\min_{b \in B} \|a - b\|_1 \leqslant \boldsymbol{D}_a \leqslant \lambda \min_{b \in B} \|a - b\|_1$, for some parameter $\lambda > 1$, then sampling $O(\lambda)$ samples results in an estimator $\hat{z}$ that is within 1% relative error to the true answer with probability at least 99%. Formally, we consider the estimator defined in Algorithm 1, where we assume access to CrudeNN$(A, B)$, a sub-routine which receives as input $A$ and $B$ and outputs estimates $\boldsymbol{D}_a \in \mathbb{R}_{\geqslant 0}$ for each $a \in A$ which is guaranteed to be an upper bound for $\min_{b \in B} \|a - b\|_1$. Based on the values $\{\boldsymbol{D}_a\}_{a \in A}$ we construct an importance sampling distribution $\mathcal{D}$ supported on $A$. As a result, we obtain the following lemma:

**Lemma 2.3** (Variance Bounds for Chamfer Estimate). *Let $n, d \in \mathbb{N}$ and suppose $A, B$ are two subsets of $\mathbb{R}^d$ of size at most $n$. For any $T \in \mathbb{N}$, the output $\boldsymbol{\eta}$ of Chamfer-Estimate$(A, B, T)$ satisfies*

$$\mathbf{E}[\boldsymbol{\eta}] = \mathrm{CH}(A, B),$$

$$\mathbf{Var}[\boldsymbol{\eta}] \leqslant \frac{1}{T} \cdot \mathrm{CH}(A, B)^2 \left(\frac{\boldsymbol{D}}{\mathrm{CH}(A, B)} - 1\right),$$

*for $\boldsymbol{D}$ from Line 1 in Figure 1. The expectations and variance are over the randomness in the samples of Line 3 of* $\texttt{Chamfer-Estimate}(A, B, T)$. *In particular,*

$$\mathbf{Pr}\left[\,|\boldsymbol{\eta} - \mathrm{CH}(A, B)| \geqslant \varepsilon \cdot \mathrm{CH}(A, B)\right] \leqslant \frac{1}{\varepsilon^2 \cdot T}\left(\frac{\boldsymbol{D}}{\mathrm{CH}(A, B)} - 1\right).$$

The proof follows from a standard analysis of importance sampling and is deferred to Appendix A. Observe, if $\boldsymbol{D} \leqslant \lambda \mathrm{CH}(A, B)$, it suffices to sample $T = O(\lambda/\varepsilon^2)$ points in $A$, leading to a running time of $O(nd\lambda/\varepsilon^2)$.

---

Subroutine $\texttt{CrudeNN}(A, B)$
**Input:** Two subsets $A, B$ of a metric space $(X, \|\cdot\|_1)$ of size at most $n$ such that all non-zero distances between any point in $A$ and any point in $B$ is between $1$ and $\mathrm{poly}(n/\varepsilon)$. We assume access to a locality-sensitive hash family at every scale $\mathcal{H}(r)$ for any $r \geqslant 0$ satisfying conditions of Definition 2.2. (We show in Appendix A that, for $\ell_1$ and $\ell_2$, the desired hash families exist, and that distances between $1$ and $\mathrm{poly}(n/\varepsilon)$ is without loss of generality).
**Output:** A list of numbers $\{\boldsymbol{D}_a\}_{a \in A}$ where $\boldsymbol{D}_a \geqslant \min_{b \in B} \|a - b\|_1$.
1. We instantiate $L = \mathcal{O}(\log(n/\varepsilon))$ and for $i \in \{0, \ldots, L\}$, we let $r_i = 2^i$.
2. For each $i \in \{0, \ldots, L\}$ sample a hash function $\boldsymbol{h}_i : X \to U$ from $\boldsymbol{h}_i \sim \mathcal{H}(r_i)$.
3. For each $a \in A$, find the smallest $i \in \{0, \ldots, L\}$ for which there exists a point $b \in B$ with $\boldsymbol{h}_i(a) = \boldsymbol{h}_i(b)$, and set $\boldsymbol{D}_a = \|a - b\|_1$.
   - The above may be done by first hashing each point $b \in B$ and $i \in \{0, \ldots, L\}$ according to $\boldsymbol{h}_i(b)$. Then, for each $a \in A$, we iterate through $i \in \{0, \ldots, L\}$ while hashing $a$ according to $\boldsymbol{h}_i(a)$ until the first $b \in B$ with $\boldsymbol{h}_i(a) = \boldsymbol{h}_i(b)$ is found.

---

Figure 2: The $\texttt{CrudeNN}$ Algorithm.

**Obtaining importance sampling probabilities.** It remains to show how to implement the $\texttt{CrudeNN}(A, B)$ subroutine to obtain the distribution over elements in $A$ which is a reasonable over-estimator of the true probabilities. A natural first step is to consider performing an $\mathcal{O}(\log n)$-approximate nearest neighbor search (NNS): for every $a' \in A$, find $b' \in B$ satisfying $\|a' - b'\|_1 / \min_{b \in B} \|a' - b\|_1 = \mathcal{O}(\log n)$. This leads to the desired guarantees on $\{\boldsymbol{D}_a\}_{a \in A}$. Unfortunately, the state of the art algorithms for $\mathcal{O}(\log n)$-approximate NNS, even under the $\ell_1$ norm, posses extraneous $\mathrm{poly}(\log n)$ factors in the runtime, resulting in a significantly higher running time. These factors are even higher for the $\ell_2$ norm. Therefore, instead of performing a direct reduction to approximate NNS, we open up the approximate NNS black-box and give a simple algorithm which directly satisfies our desired guarantees on $\{\boldsymbol{D}_a\}_{a \in A}$.

To begin with, we assume that the aspect ratio of all pair-wise distances is bounded by a fixed polynomial, $\mathrm{poly}(n/\varepsilon)$ (we defer the reduction from an arbitrary input to one with polynomially bounded aspect ratio to Lemma A.3). We proceed via computing $\mathcal{O}(\log(n/\varepsilon))$ different (randomized) partitions of the dataset $A \cup B$. The $i$-th partition, for $1 \leqslant i \leqslant \mathcal{O}(\log(n/\varepsilon))$, can be written as $A \cup B = \cup_j \mathcal{P}_j^i$ and approximately satisfies the property that points in $A \cup B$ that are at distance at most $2^i$ will be in the same partition $\mathcal{P}_j^i$ with sufficiently large probability. To obtain these components, we use a family of *locality-sensitive hash functions*, whose formal properties are given in Definition 2.2. Intuitively, these hash functions guarantee that:

1. For each $a' \in A$, its *true* nearest neighbor $b' \in B$ falls into the *same* component as $a'$ in the $i_0$-th partition, where $2^{i_0} = \Theta(\|a' - b'\|_1)$ [2], and

2. Every other extraneous $b \neq b'$ is *not* in the same component as $a'$ for each $i < i_0$.

It is easy to check that any hash function that satisfies the aforementioned guarantees yields a valid set of distances $\{\boldsymbol{D}_a\}_{a \in A}$ as follows: for every $a' \in A$, find the smallest $i_0$ for which there exists a $b' \in B$ in the same component as $a'$ in the $i_0$-th partition. Then set $\boldsymbol{D}_{a'} = \|a' - b'\|_1$. Intuitively, the $b'$ we find for any fixed $a'$ in this procedure will have distance that is at least the closest neighbor in $B$ and with good probability, it won't be too much larger. A caveat here is that we cannot show the

---

[2]Recall we assumed all distances are between $1$ and $\mathrm{poly}(n)$ resulting in only $\mathcal{O}(\log n)$ different partitions

above guarantee holds for $2^{i_0} = \Theta(\|a' - b'\|_1)$. Instead, we obtain the slightly weaker guarantee that, *in the expectation*, the partition $b'$ lands in is a $\mathcal{O}(\log n)$-approximation to the minimum distance, i.e. $2^{i_0} = \Theta(\log n \cdot \|a' - b'\|_1)$. Therefore, after running $\mathtt{CrudeNN}(A, B)$, setting $\lambda = \log n$ suffices for our $\mathcal{O}(nd\log(n)/\varepsilon^2)$ time algorithm. We formalize this argument in the following lemma:

**Lemma 2.4** (Oversampling with bounded Aspect Ratio). *Let $(X, d_X)$ be a metric space with a locality-sensitive hash family at every scale (see Definition 2.2). Consider two subsets $A, B \subset X$ of size at most $n$ and any $\varepsilon \in (0, 1)$ satisfying*

$$1 \leqslant \min_{\substack{a \in A, b \in B \\ a \neq b}} d_X(a, b) \leqslant \max_{a \in A, b \in B} d_X(a, b) \leqslant \mathrm{poly}(n/\varepsilon).$$

*Algorithm 2, $\mathtt{CrudeNN}(A, B)$, outputs a list of (random) positive numbers $\{\boldsymbol{D}_a\}_{a \in A}$ which satisfy the following two guarantees:*

- *With probability $1$, every $a \in A$ satisfies $\boldsymbol{D}_a \geqslant \min_{b \in B} d_X(a, b)$.*

- *For every $a \in A$, $\mathbf{E}[\boldsymbol{D}_a] \leqslant \mathcal{O}(\log n) \cdot \min_{b \in B} d_X(a, b)$.*

*Further, Algorithm 2, runs in time $\mathcal{O}(dn\log(n/\varepsilon))$ time, assuming that each function used in the algorithm can be evaluated in $\mathcal{O}(d)$ time.*

*Proof Sketch for Theorem 2.1.* Given the lemmas above, it is straight-forward to complete the proof of Theorem 2.1. First, we reduce to the setting where the aspect ratio is $\mathrm{poly}(n/\varepsilon)$ (see Lemma A.3 for a formal reduction). We then invoke Lemma 2.4 and apply Markov's inequality to obtain a set of distances $\boldsymbol{D}_a$ such that with probability at least $99/100$, for each $a \in A$, $\min_{b \in B} \|a - b\|_1 \leqslant \boldsymbol{D}_a$ and $\sum_{a \in A} \boldsymbol{D}_a \leqslant \mathcal{O}(\log(n)) \, \mathtt{CH}(A, B)$. We then invoke Lemma 2.3 and set the number of samples, $T = \mathcal{O}(\log(n)/\varepsilon^2)$. The running time of our algorithm is then given by the time of $\mathtt{CrudeNN}(A, B)$, which is $O(nd\log(n/\varepsilon))$, and the time needed to evaluate the estimator in Lemma 2.3, requiring $\mathcal{O}(nd\log(n)/\varepsilon^2)$ time. Refer to Section A for the full proof. $\qquad\square$

**Other Related Works** We note that importance sampling is a popular technique used for speeding up geometric algorithms. For example, [14] uses it to obtain a fast $c$-approximate algorithm for computing Earth Mover Distance (EMD) in two (or any constant) dimensions, for some constant $c > 2$. However, the application and implementation of importance sampling in that paper is quite different from ours. In [14], the space containing all input points is subdivided into regions, and the total EMD value is represented as a sum of EMDs restricted to point-sets in each region (plus an additional representing the "global" EMD). The EMD cost in each region is then approximated quickly by embedding EMD into $\ell_1$ using a randomly shifted quadtree with logarithmic distortion; these estimations define the sampling probabilities.

In contrast, in our paper, the value of the Chamfer distance is exactly equal to the sum of distances from each point to its nearest neighbor, so there is no decomposition involved. Instead, we approximate each distance to nearest neighbor using a randomized hierarchical decomposition; for the case of the $\ell_1$ norm, each level of the decomposition partitions the space into rectangular boxes, as in quadtrees. Crucially, however, to ensure that the running time of our algorithm is within the stated bounds, we cannot use a standard randomly shifted quadtree where each level is shifted by the same random vector (as in [14]). This is because shifting all levels by the same amount only ensures that the expected distortion between a fixed pair of points is logarithmic; to ensure that the distance to the nearest neighbor is distorted by $O(\log n)$, we would need to use $O(\log n)$ independent quadtrees and apply the union bound. Instead, we use independent random partitions at each level, and show (Lemma 2.4) that this suffices to bound the expected distortion of the distance to the nearest neighbor, without incurring any additional factors. This makes it possible to obtain the running time as stated.

There are other works on quickly computing EMD and related distances. For example, the algorithm of [4] runs in time that is linear in the number of distances, i.e. it runs in $\Omega(n^2)$ time and gets a $1 + \varepsilon$ approximation to EMD. This means that their approach requires a runtime quadratic in the size of the dataset $n$. In contrast, Chamfer distance admits a trivial time $O(n^2)$ algorithm and our main contribution is to provide a nearly linear $O(n\log(n)/\varepsilon^2)$ algorithm to get $(1 + \varepsilon)$-approximation to Chamfer distance.

# 3 Experiments

We perform an empirical evaluation of our Chamfer distance estimation algorithm.

**Summary of Results**   Our experiments demonstrate the effectiveness of our algorithm for both low and high dimensional datasets and across different dataset sizes. Overall, it is much faster than brute force (even accelerated with KD-trees). Further, our algorithm is both faster and more sample-efficient than uniform sampling. It is also robust to different datasets: while uniform sampling performs well for most datasets in our experiments, it performs poorly on datasets where the distances from points in $A$ to their neighbors in $B$ vary significantly. In such cases, our algorithm is able to adapt its importance sampling probabilities appropriately and obtain significant improvements over uniform sampling.

| Dataset | $|A|, |B|$ | $d$ | Experiment | Metric | Reference |
|---|---|---|---|---|---|
| ShapeNet | $\sim 8 \cdot 10^3, \sim 8 \cdot 10^3$ | 3 | Small Scale | $\ell_1$ | [10] |
| Text Embeddings | $2.5 \cdot 10^3, 1.8 \cdot 10^3$ | 300 | Small Scale | $\ell_1$ | [16] |
| Gaussian Points | $5 \cdot 10^4, 5 \cdot 10^4$ | 2 | Outliers | $\ell_1$ | - |
| DEEP1B | $10^4, 10^9$ | 96 | Large Scale | $\ell_2$ | [9] |
| Microsoft-Turing | $10^5, 10^9$ | 100 | Large Scale | $\ell_2$ | [21] |

Table 1: Summary of our datasets. For ShapeNet, the value of $|A|$ and $|B|$ is averaged across different point clouds in the dataset.

## 3.1 Experimental Setup

We use three different experimental setups, small scale, outlier, and large scale. They are designed to 'stress test' our algorithm, and relevant baselines, under vastly different parameter regimes. The datasets we use are summarized in Table 1. For all experiments, we introduce uniform sampling as a competitive baseline for estimating the Chamfer distance, as well as (accelerated) brute force computation. All results are averaged across 20+ trials and 1 standard deviation error bars are shown when relevant.

**Small Scale**   These experiments are motivated from common use cases of Chamfer distance in the computer vision and NLP domains. In our small scale experiments, we use two different datasets: (a) the ShapeNet dataset, a collection of point clouds of objects in three dimensions [10]. ShapeNet is a common benchmark dataset frequently used in computer graphics, computer vision, robotics and Chamfer distance is a widely used measure of similarity between different ShapeNet point clouds [10]. (b) We create point clouds of words from text documents from [16]. Each point represents a word embedding obtained from the word-to-vec model of [19] in $\mathbb{R}^{300}$ applied to the Federalist Papers corpus. As mentioned earlier, a popular relaxation of the common Earth Mover Distance is exactly the (weighted) version of the Chamfer distance [16, 7].

Since ShapenNet is in three dimensions, we implement nearest neighbor queries using KD-trees to accelerate the brute force baseline as KD-trees can perform exact nearest neighbor search quickly in small dimensions. However, they have runtime exponential in dimension meaning they cannot be used for the text embedding dataset, for which we use a standard naive brute force computation. For both these datasets, we implement our algorithms using Python 3.9.7 on an M1 MacbookPro with 32GB of RAM. We also use an efficient implementation of KD trees in Python and use Numpy and Numba whenever relevant. Since the point clouds in the dataset have approximately the same $n$ value, we compute the symmetric version $\text{CH}(A, B) + \text{CH}(B, A)$. For these experiments, we use the $\ell_1$ distance function.

**Outliers**   This experiment is meant to showcase the robustness of our algorithm. We consider two point clouds, $A$ and $B$, each sampled from Gaussian points in $\mathbb{R}^{100}$ with identity covariance. Furthermore, we add an "outlier" point to $A$ equal to $0.5n \cdot \mathbf{1}$, where $\mathbf{1}$ is the all ones vector.

This example models scenarios where the distances from points in $A$ to their nearest neighbors in $B$ vary significantly, and thus uniform sampling might not accurately account for all distances, missing a small fraction of large ones.

**Large Scale** The purpose of these experiments is to demonstrate that our method scales to datasets with billions of points in hundreds of dimensions. We use two challenging approximate nearest neighbor search datasets: DEEP1B [9] and Microsoft Turing-ANNS [21]. For these datasets, the set $A$ is the query data associated with the datasets. Due to the asymmetric sizes, we compute $\text{CH}(A, B)$. These datasets are normalized to have unit norm and we consider the $\ell_2$ distance function.

These datasets are too large to handle using the prior configurations. Thus, we use a proprietary in-memory parallel implementation of the SimHash algorithm, which is an $\ell_2$ LSH family for normalized vectors according to Definition 2.2 [11], on a shared virtual compute cluster with 2x64 core AMD Epyc 7763 CPUs (Zen3) with 2.45Ghz - 3.5GHz clock frequency, 2TB DDR4 RAM and 256 MB L3 cache. We also utilize parallization on the same compute cluster for naive brute force search.

## 3.2 Results

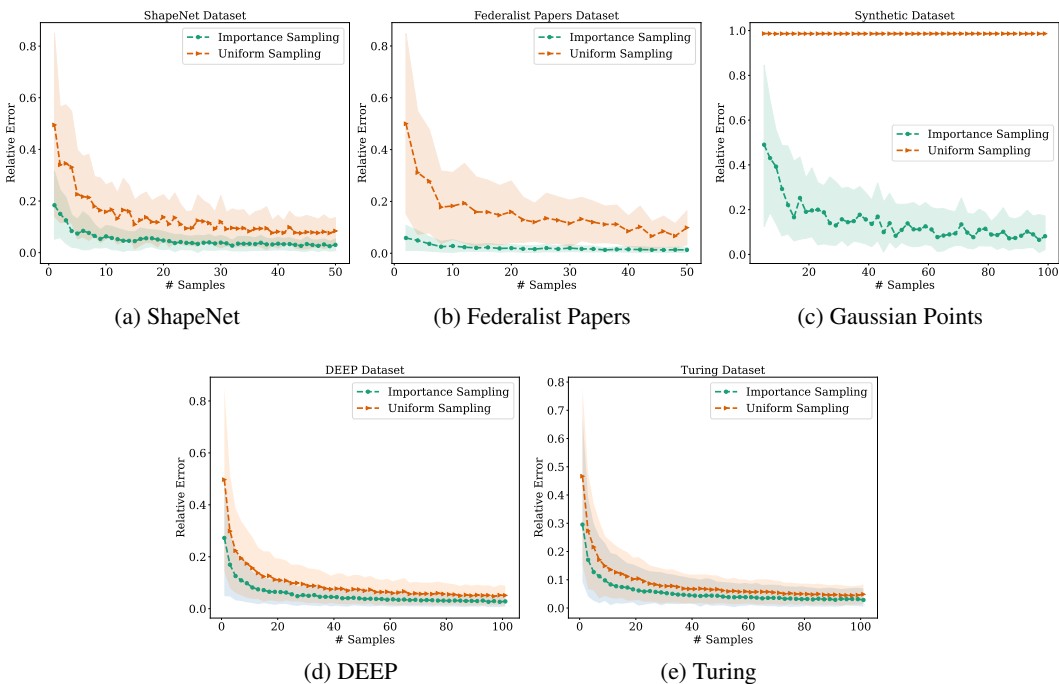

Figure 3: Sample complexity vs relative error curves.

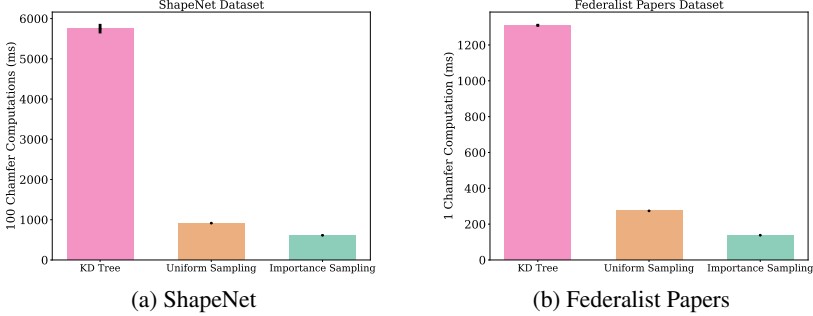

Figure 4: Runtime experiments. We set the number of samples for uniform and importance sampling such that the relative errors of their respective approximations are similar.

**Small Scale** First we discuss configuring parameters. Recall that in our theoretical results, we use $\mathcal{O}(\log n)$ different scales of the LSH family in $\texttt{CrudeNN}$. $\texttt{CrudeNN}$ then computes (over) estimates of the nearest neighbor distance from points in $A$ to $B$ (in near linear time) which is then used

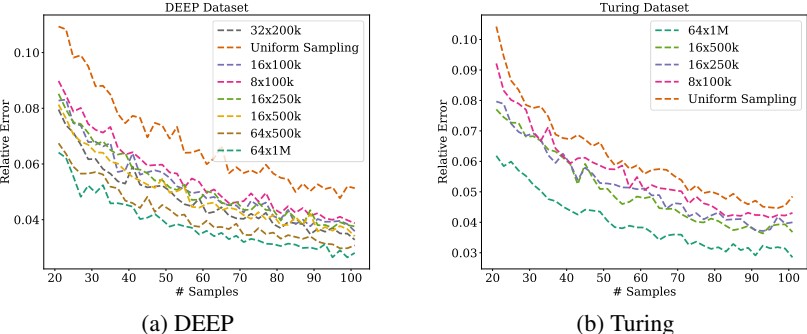

(a) DEEP                                    (b) Turing

Figure 5: The figures show sample complexity vs relative error curves as we vary the number of LSH data structures and window sizes. Each curve maps $k \times W$ where $k$ is the number of LSH data structures we use to repeatedly hash points in $B$ and $W$ is the window size, the number of points retrieved from $B$ that hash closest to any given $a$ at the smallest possible distance scales.

for importance sampling by `Chamfer-Estimate`. Concretely for the $\ell_1$ case, this the LSH family corresponds to imposing $\mathcal{O}(\log n)$ grids with progressively smaller side lengths. In our experiments, we treat the number of levels of grids to use as a tuneable parameter in our implementation and find that a very small number suffices for high quality results in the importance sampling phase.

Figure 6 (b) shows that only using 3 grid levels is sufficient for the crude estimates $\boldsymbol{D}_a$ to be within a factor of 2 away from the true nearest neighbor values for the ShapeNet dataset, averaged across different point clouds in the dataset. Thus for the rest of the Small Scale experiments, we fix the number of grid levels to be 3.

Figure 3 (a) shows the sample complexity vs accuracy trade offs of our algorithm, which uses importance sampling, compared to uniform sampling. Accuracy is measured by the relative error to the true value. We see that our algorithm possesses a better trade off as we obtain the same relative error using only 10 samples as uniform sampling does using $50+$ samples, resulting in at least a **5x** improvement in sample complexity. For the text embedding dataset, the performance gap between our importance sampling algorithm and uniform sampling grows even wider, as demonstrated by Figure 3 (b), leading to **> 10x** improvement in sample complexity.

In terms of runtimes, we expect the brute force search to be much slower than either importance sampling and uniform sampling. Furthermore, our algorithm has the overhead of first estimating the values $\boldsymbol{D}_a$ for $a \in A$ using an LSH family, which uniform sampling does not. However, this is compensated by the fact that our algorithm requires much fewer samples to get accurate estimates.

Indeed, Figure 4 (a) shows the average time of 100 Chamfer distance computations between randomly chosen pairs of point clouds in the ShapeNet dataset. We set the number of samples for uniform sampling and importance sampling (our algorithm) such that they both output estimates with (close to) $2\%$ relative error. Note that our runtime includes the time to build our LSH data structures. This means we used 100 samples for importance sampling and 500 for uniform. The brute force KD Tree algorithm (which reports exact answers) is approximately 5x slower than our algorithm. At the same time, our algorithm is $50\%$ faster than uniform sampling. For the Federalist Papers dataset (Figure 4 (b)), our algorithm only required 20 samples to get a $2\%$ relative error approximation, whereas uniform sampling required at least 450 samples. As a result, our algorithm achieved **2x** speedup compared to uniform sampling.

**Outliers** We performed similar experiments as above. Figure 3 (c) shows the sample complexity vs accuracy trade off curves of our algorithm and uniform sampling. Uniform sampling has a very large error compared to our algorithm, as expected. While the relative error of our algorithm decreases smoothly as the sample size grows, uniform sampling has the same high relative error. In fact, the relative error will stay high until the outlier is sampled, which typically requires $\Omega(n)$ samples.

**Large Scale** We consider two modifications to our algorithm to optimize the performance of `CrudeNN` on the two challenging datasets that we are using; namely, note that both datasets are standard for benchmarking billion-scale nearest neighbor search. First, in the `CrudeNN` algorithm, when computing $\boldsymbol{D}_a$ for $a \in A$, we search through the hash buckets $h_1(a), h_2(a), \dots$ containing

$a$ in increasing order of $i$ (i.e., smallest scale first), and retrieve the first $W$ (window size) distinct points in $B$ from these buckets. Then, the whole process is repeated $k$ times, with $k$ independent LSH data structures, and $\boldsymbol{D}_a$ is set to be the distance from $a$ to the closest among all $Wk$ retrieved points.

Note that previously, for our smaller datasets, we set $\boldsymbol{D}_a$ to be the distance to the first point in $B$ colliding with $a$, and repeated the LSH data structure once, corresponding to $W = k = 1$. In our figures, we refer to these parameter choices as $k \times W$ and test our algorithm across several choices.

For the DEEP and Turing datasets, Figures 3 (d) and 3 (e) show the sample complexity vs relative error trade-offs for the best parameter choice (both $64 \times 10^6$) compared to uniform sampling. Qualitatively, we observe the same behavior as before: importance sampling requires fewer samples to obtain the same accuracy as uniform sampling. Regarding the other parameter choices, we see that, as expected, if we decrease $k$ (the number of LSH data structures), or if we decrease $W$ (the window size), the quality of the approximations $\{\boldsymbol{D}_a\}_{a \in A}$ decreases and importance sampling has worse sample complexity trade-offs. Nevertheless, for all parameter choices, we see that we obtain superior sample complexity trade-offs compared to uniform sampling, as shown in Figure 5. A difference between these parameter choices are the runtimes required to construct the approximations $\{\boldsymbol{D}_a\}_{a \in A}$. For example for the DEEP dataset, the naive brute force approach (which is also optimized using parallelization) took approximately $1.3 \cdot 10^4$ seconds, whereas the most expensive parameter choice of $64 \times 10^6$ took approximately half the time at $6.4 \times 10^3$ and the cheapest parameter choice of $8 \times 10^5$ took 225 seconds, leading to a **2x-50x** factor speedup. The runtime differences between brute force and our algorithm were qualitative similar for the Turing dataset.

Similar to the small scale dataset, our method also outperforms uniform sampling in terms of runtime if we require they both output high quality approximations. If we measure the runtime to get a $1\%$ relative error, the $16 \times 2 \cdot 10^5$ version of our algorithm for the DEEP dataset requires approximately 980 samples with total runtime approximately 1785 seconds, whereas uniform sampling requires $> 1750$ samples and runtime $> 2200$ seconds, which is $> 23\%$ slower. The gap in runtime increases if we desire approximations with even smaller relative error, as the overhead of obtaining the approximations $\{\boldsymbol{D}_a\}_{a \in A}$ becomes increasingly overwhelmed by the time needed to compute the exact answer for our samples.

**Additional Experimental Results**    We perform additional experiments to show the utility of our approximation algorithm for the Chamfer distance for downstream tasks. For the ShapeNet dataset, we show we can efficiently recover the true exact nearest neighbor of a fixed point cloud $A$ in Chamfer distance among a large collect of different point clouds. In other words, it is beneficial for finding the 'nearest neighboring point cloud'. Recall the ShapeNet dataset, contains approximately $5 \cdot 10^4$ different point clouds. We consider the following simple (and standard) two step pipeline: (1) use our algorithm to compute an approximation of the Chamfer distance from $A$ to every other point cloud $B$ in our dataset. More specifically, compute an approximation to $\mathrm{CH}(A, B) + \mathrm{CH}(B, A)$ for all $B$ using 50 samples and the same parameter configurations as the small scale experiments. Then filter the dataset of points clouds and prune down to the top $k$ closest point cloud candidates according to our approximate distances. (2) Find the closest point cloud in the top $k$ candidates via exact computation.

We measure the accuracy of this via the standard recall $@k$ measure, which computes the fraction of times the *exact* nearest neighbor $B$ of $A$, averaged over multiple $A$'s, is within the top $k$ choices. Figure 6 (a) shows that the true exact nearest neighbor of $A$, that is the point cloud $B$ which minimizes $\mathrm{CH}(A, B) + \mathrm{CH}(B, A)$ among our collection of multiple point clouds, is within the top 30 candidates $> 98\%$, time (averaged over multiple different choices of $A$). This represents a more than **1000x** reduction in the number of point clouds we do exact computation over compared to the naive brute force method, demonstrating the utility of our algorithm for downstream tasks.

## 4    Lower Bound for Reporting the Alignment

We presented an algorithm that, in time $\mathcal{O}\big(nd \log(n)/\varepsilon^2\big)$, produces a $(1 + \varepsilon)$-approximation to $\mathrm{CH}(A, B)$. It is natural to ask whether it is also possible to *report* a mapping $g : A \to B$ whose cost $\sum_{a \in A} \|a - g(a)\|_1$ is within a factor of $1 + \varepsilon$ from $\mathrm{CH}(A, B)$. (Our algorithm uses on random sampling and thusdoes not give such a mapping). This section shows that, under a popular complexity-theoretic conjecture called the *Hitting Set Conjecture* [24], such an algorithm does not exists. For simplicity, we focus on the case when the underlying metric $d_X$ is induced by the Manhattan distance,

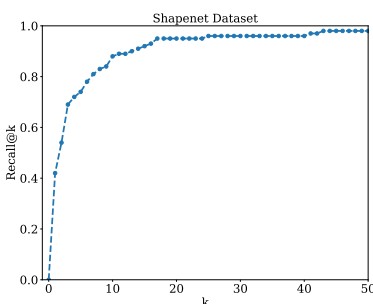
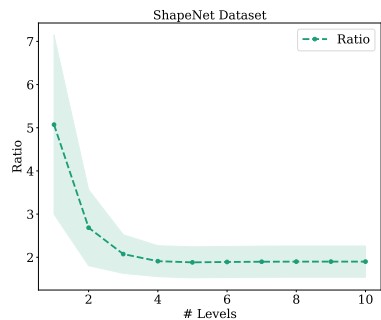

(a) ShapeNet NNS pipeline experiments

(b) Quality of approximations $\boldsymbol{D}_a$ vs the number of levels of LSH data structure

Figure 6: Additional figures for the ShapeNet dataset.

i.e., $d_X(a, b) = \|a - b\|_1$. The argument is similar for the Euclidean distance, Euclidean distance squared, etc. To state our result formally, we first define the Hitting Set (HS) problem.

**Definition 4.1** (Hitting Set (HS) problem). The input to the problem consists of two sets of vectors $A, B \subseteq \{0, 1\}^d$, and the goal is to determine whether there exists some $a \in A$ such that $a \cdot b \neq 0$ for every $b \in B$. If such an $a \in A$ exists, we say that $a$ *hits* $B$.

It is easy to see that the Hitting Set problem can be solved in time $\mathcal{O}(n^2 d)$. The Hitting Set Conjecture [24] postulates that this running time is close to the optimal. Specifically:

**Conjecture 4.2.** *Suppose $d = \Theta(\log^2 n)$. Then for every constant $\delta > 0$, no randomized algorithm can solve the Hitting Set problem in $\mathcal{O}(n^{2-\delta})$ time.*

Our result can be now phrased as follows.

**Theorem 4.3** (Hardness for reporting a mapping). *Let $T(N, D, \varepsilon)$ be the running time of an algorithm ALG that, given sets of $A", B" \subset \{0, 1\}^D$ of sizes at most $N$, reports a mapping $g : A" \to B"$ with cost $(1 + \varepsilon)\mathrm{CH}(A", B")$, for $D = \Theta(\log^2 N)$ and $\varepsilon = \frac{\Theta(1)}{D}$. Assuming the Hitting Set Conjecture, we have that $T(N, D, \varepsilon)$ is at least $\Omega(N^{2-\delta})$ for any constant $\delta > 0$.*

## 5  Conclusion

We present an efficient approximation algorithm for estimating the Chamfer distance up to a $1 + \varepsilon$ factor in time $\mathcal{O}(nd \log(n)/\varepsilon^2)$. The result is complemented with a conditional lower bound which shows that reporting a Chamfer distance mapping of similar quality requires nearly quadratic time. Our algorithm is easy to implement in practice and compares favorably to brute force computation and uniform sampling. We envision our main tools of obtaining fast estimates of coarse nearest neighbor distances combined with importance sampling can have additional applications in the analysis of high-dimensional, large scale data.

## 6  Acknowledgement

AB was supported by Ankur Moitra's ONR grant and the NSF TRIPODS program (award DMS-2022448). PI was supported by the NSF TRIPODS program (award DMS-2022448), Simons Investigator Award and GIST-MIT Research Collaboration grant.

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
