# OpenReview forum: "Near-Linear Time Algorithm for the Chamfer Distance"
_NeurIPS.cc/2023/Conference — NeurIPS 2023 poster_

### Official Review · Reviewer_5NRb · 2023-06-09

**Soundness:** 3 good
**Presentation:** 2 fair
**Contribution:** 3 good
**Rating:** 5
**Confidence:** 3

**Summary:**

The paper proposes a fast estimation of Chamfer distance which is a famous distance between two sets. The key idea is to use random sampling on the first set and then find the closest point on the other set. To enhance further the approximation performance and quality of samples, importance sampling is utilized. The paper proposes to use a proposal $D_a$ such that for any $a \in A$, we have $ min_{b \in B} |a-b| \leq D_a \leq \lambda  min_{b\in B} |a-b| $ for some $\lambda >1$.  From that, the paper can obtain an $\epsilon$-approximation computational complexity of the estimation. To obtain the proposal distribution, a fast sub-routine (CrudeNN) is proposed. The experiments are designed to show that the proposed importance sampling has better relative error than the uniform sampling and is faster than the  KD-tree implementation.

**Strengths:**

* Stochastic approximation of Chamfer distance is new.
* The design of the importance sampling proposal is interesting.
* An approximation error can be obtained.
* The final approximation is faster than the conventional Chamfer distance.

**Weaknesses:**

* The proposed algorithm can only work with some restricted set of ground metrics that satisfies  (Hashing at every scale).
* The main weakness of the paper is lacking real applications comparison. For example, the paper can compare the proposed approximation with the conventional Chamfer distance on a point-cloud reconstruction/completion task.
* In deep learning applications, the gradient of Chamfer distance with respect to an input set is required. Therefore, some investigation about the approximation error of the gradient should be considered.


**Questions:**

* As the paper mentions Chamfer is a fast alternative distance to EMD, the stochastic approximation of EMD should be discussed [1].

[1] Improving Mini-batch Optimal Transport via Partial Transportation

* For near-linear computational complexity, the sliced Wasserstein distance has the same complexity. Therefore, it is a direct competitor to the proposed approximated Chamfer and should be compared in a benchmark point-cloud task in the paper.

**Limitations:**

Not applicable.

---

> ### Author Rebuttal · Authors · 2023-08-08
>
> We thank the reviewer for their detailed and thorough comments. See our answers to specific comments below.
>
> **On the set of ground metrics which the algorithm works for.**
>
> We emphasize that our methods work for any metrics which support a locality sensitive hash function (or such metrics squared). This encompasses a very wide class of metrics including $\ell_1, \ell_2$ (and generally all $\ell_p$ for $1\le p \le 2$, the Jaccard distance, and even includes functions which are not strictly metrics such as $\ell_p$ for $0 < p < 1$ and $\ell_2^2$.  In addition, using standard metric embedding theory allows us to extend our results to more general metrics. We believe this demonstrates the wide applicability of our methods.
>
> **On real application comparisons**
>
> We agree with the reviewer that the Chamfer distance is widely used in point-cloud reconstruction and completion tasks, often as a component of the loss function in a deep network. However, we would like to point out that the Chamfer distance is also widely used in many use cases beyond these tasks, with a very common application being comparing the similarity between two point clouds. This is evidenced by the fact that the Chamfer distance is implemented in many popular libraries such as PDAL [1], CloudCompare [2], PointCloud Utils [3], for the purposes of comparing point clouds.
> The main practical application of our theoretical results is indeed to speed up the chamfer distance for point cloud comparisons with strong provable guarantees. Our experiments complement our theory and demonstrate its effectiveness, even on billion scale datasets.
> Exploring the applicability of our method for reconstruction and completion tasks, which will likely require efficient estimation of the gradient of the chamfer distance, is an interesting future research direction.
>
> [1] PDAL, Documentation » Applications » chamfer
>
> [2] CloudCompare, User’s Manual for version 2.1
>
> [3] Point Cloud Utils, Functionality » Point Cloud Metrics​
>
> **Investigation on the approximation error of the gradient.**
>
> Thank you for the comment. As stated above, we agree that exploring the application of methods to computing the gradient of the chamfer distance is an interesting future direction.
>
>
> "**The stochastic approximation of EMD should be discussed Improving Mini-batch Optimal Transport via Partial Transportation**"
>
> Thank you for the reference. We would like to point out that the focus of our paper is not to motivate or argue for why the Chamfer distance is a useful approximation to the EMD, or an interesting similarity measure in its own right. This is a well established notion in the ML literature (as evidenced by the references in our paper and the references above).
>
> With regards to the proposed reference, it is our understanding that while the proposed algorithm is fast, it cannot effectively approximate the EMD. The reference proposes computing the EMD on a smaller subsampled set of points as a proxy for the full computation on the entire dataset. In the worst case, this approach can require sampling almost all of the dataset to form any accurate approximation of the EMD, offering no practical speed up over the naive computation. To see this, consider the case where the two point sets A and B are both of size n and each have n-1 points at the origin. The nth point of A is located at 10 on the real line and the nth point of B is located at 11. The true EMD distance between A and B is clearly 1. However, if we don’t subsample Omega(n) points, then we will not sample the nth points of A and B, implying that the computed estimate is 0 with high probability. While this exact example is unlikely to occur in practice, it shows the fragility of the proposed method. In contrast, we have provable worst case guarantees for computing the chamfer distance in near linear time up to arbitrary accurate approximation, which we believe is a valuable contribution to the ML community.
>
>
> **On the sliced Wasserstein distance.**
>
> Thank you for the reference. We agree that the sliced Wasserstein distance has a linear complexity, *if* the number of projections used is constant. However in practice, a prohibitively large number of projections are commonly used to effectively estimate the sliced Wassertein distance. For example, many practical works (see [a,b,c,d] below) recommend using at least d different random projections, where d is the dimension of the dataset, leading to a runtime of at least Omega(nd^2). In contrast, our method for the Chamfer distance provably runs in O(nd log n) time, to provide a 1% relative error estimate for worst case inputs.
>
> [a] Hierarchical Sliced Wasserstein Distance. Khai Nguyen, Tongzheng Ren, Huy Nguyen, Litu Rout, Tan Nguyen, Nhat Ho. 2023
>
> [b] Distributional sliced-Wasserstein and applications to generative modeling. Khai Nguyen, Nhat Ho, Tung Pham, and Hung Bui. 2021.
>
> [c] Improving relational regularized autoencoders with spherical sliced fused Gromov-Wasserstein. Khai Nguyen, Son Nguyen, Nhat Ho, Tung Pham, and Hung Bui. 2021.
>
> [d] Generative modeling using the sliced Wasserstein distance. Ishan Deshpande, Ziyu Zhang, Alexander G Schwing. 2018

---

> > ### Comment · Reviewer_5NRb · 2023-08-11
> > **Response to the authors**
> >
> > Thank you for your response.
> >
> > I acknowledge the rebuttal of the authors which helps me to understand clearly the contribution of the paper. Based on the response, I raised my score to 5. However, I reduced my confidence to 3 since I think I did not understand correctly the contribution of the paper in the first round. Overall, I believe the discussion with EMD-related metrics could benefit the impact of the paper in applications.
> >
> > Best regards

---

### Official Review · Reviewer_qK6W · 2023-07-07

**Soundness:** 4 excellent
**Presentation:** 3 good
**Contribution:** 3 good
**Rating:** 6
**Confidence:** 4

**Summary:**

This paper provides the first near-linear time approximation scheme to compute the Chamfer distance $\texttt{CH}(A,B) = \sum_{a \in A} \min_{b \in B} d_X(a,b)$ for any two point sets $A,B \in \mathbb{R}^d$ and $d_X$ is the distance measure under consideration. In particular, for $|A|, |B| \leq n$, and for any $\epsilon > 0$, they provide an algorithm which outputs a $(1 + \epsilon)$-approximate solution with probability greater than 99% in time $O(nd \log(n)/\epsilon^2)$. They also provide experimental evaluation of their algorithm and also a matching lower bound for the harder task of not just computing the value of the Chamfer distance but also providing a valid mapping from $A \mapsto B$ which achieves that Chamfer distance.

**Strengths:**

Originality: The paper is the first to provide a near-linear time approximation scheme for computing Chamfer distances.

Quality and Clarity: Paper is overall pretty well-written.

Significance: I have personally not used Chamfer distances in any of my prior work before so cannot directly attest the importance of Chamfer distances but the authors have done a reasonably good job of motivating the problem in the introduction of the paper. It looks like Chamfer distances have already been implemented in several popular ML libraries like TensorFlow and PyTorch. They have also been used in prior papers which have appeared in NeurIPS and other related conferences.

**Weaknesses:**

Not any that I can think of right now.

**Questions:**

Not a major question, but more of a suggestion. As a person who has not worked with Chamfer distances earlier, I don't know if they are useful other than as a proxy for Earth-mover distance (at least that's what I got from your introduction). A quick google search on approximating Earth-mover distances led me to this following paper: Near-linear time approximation algorithms for optimal transport via Sinkhorn iteration, by Altschuler et.al, which appeared in NeurIPS 2017. The paper seems to be pretty well-known by now, with over 500 citations. Given that your paper is about approximating Chamfer distances, which seem to only be a proxy for Earth-mover distance, I think it is very natural that you at least refer this paper, or even better give a more detailed explanation of why your paper is relevant even though there is an approximation scheme for Earth-mover distances already.

**Limitations:**

Mainly theoretical paper.

---

> ### Author Rebuttal · Authors · 2023-08-07
>
> We thank the reviewer for their detailed and thorough comments. See our answers to specific comments below.
>
> **Applications of Chamfer Distance.**  In addition to being a proxy for earth-mover distance, Chamfer distance is widely and independently used in point-cloud reconstruction and completion tasks, often as a component of the loss function in a deep network.Chamfer distance is also widely used in many use cases beyond these tasks, with a very common application being comparing the similarity between two point clouds. This is evidenced by the fact that the Chamfer distance is implemented in many popular libraries such as PDAL [1], CloudCompare [2], PointCloud Utils [3], for the purposes of comparing point clouds.
>
> **Comparison to Altschuler et. al.** We thank the reviewer for pointing us to Altschuler et. al. The algorithm introduced by Altschuler et. al. runs in time that is linear in the number of edges in the graph, i.e. it runs in $O(n^2/\epsilon^3)$ time and gets a 1+eps approximation to EMD. This means that their approach requires a runtime quadratic in the size of the dataset n. In contrast, Chamfer distance admits a trivial $O(n^2)$ time algorithm and our main contribution is to provide a nearly *linear* $O(n \log(n)/\epsilon^2)$ algorithm to get a $(1+\epsilon)$-approximation to Chamfer distance. We will add the reference and comparison to our paper.
>
> **Empirical evaluations.** Lastly, with regards to the comment about being mainly a theoretical paper, we would like to emphasize the detailed experimental section in our paper, where we evaluated our algorithm and compared it to the brute force and uniform sampling baselines. Moreover, we evaluated on a variety of dataset types and scales, including the ShapeNet dataset of 3D point clouds, word embedding datasets, and two different billion-scale embedding datasets from Big-ANN-Benchmarks, which is a central benchmark for large-scale nearest neighbor search. Thus, our experimental section empirically demonstrates that our algorithm is applicable and performs well for a wide range of data types and scales.
>
>
>
> [1] PDAL, Documentation » Applications » chamfer
>
> [2] CloudCompare, User’s Manual for version 2.1
>
> [3] Point Cloud Utils, Functionality » Point Cloud Metrics​

---

### Official Review · Reviewer_3E3v · 2023-07-25

**Soundness:** 3 good
**Presentation:** 3 good
**Contribution:** 2 fair
**Rating:** 7
**Confidence:** 4

**Summary:**

The paper gives an algorithm to approximate Chamfer distance between two sets within $(1 \pm \epsilon) $ of the original value. The brute force algorithm to calculate the distance exactly between two sets of size $n$ and with points in $d$-dimension is $O(n^2d)$ which is prohibitive for large values of $n$. The authors use a crude version of the locality sensitive hashing technique, to design an importance sampling algorithm that samples $O(\log{n})$ points from the first set and approximates Chamfer distance only using these weighted and sampled points, hence giving an approximation algorithm with time complexity $O(nd\log{n})$. They validate the efficiency of their algorithm for a variety of datasets. They also show negative result where they show that it is not possible to give the actual mapping between the two sets which gives the $(1 + \epsilon) $approximate Chamfer distance with a faster algorithm under the assumption that the hitting set conjecture is true.

**Strengths:**

1) The paper is written very nicely and is coherent and easy to follow
2) Experiments are performed under a variety of settings and code is provided. Authors should make it public if the paper gets accepted.
3) Proofs appear sound to the best of my understanding.

Overall, the paper does a good job of efficiently approximation a quantity which finds application in a variety of problems in machine learning and will be of interest to the community.

**Weaknesses:**

1) I think it would be good if the authors can at least give the definition of Locality Sensitivity Hashing in the main body of the paper. It would increase the readability.
2) The only other arguable weakness I find is that the ideas used in the paper and the proof techniques are not novel and as such may not be so interesting from theoretical perspective.

**Questions:**

Please refer to weaknesses section

**Limitations:**

Please refer to weaknesses section

---

> ### Author Rebuttal · Authors · 2023-08-07
>
> We thank the reviewer for their detailed and thorough comments. See our answers to specific comments below.
>
> **Experiments are performed under a variety of settings and code is provided.**
> Thank you for the comment. We intend to make the code public if the paper is accepted.
>
> **Definition of Locality Sensitive Hashing.**
> Thank you for the suggestion. We will make this change to include a definition in the main body in the updated version of the paper.
>
>
> **On the novelty of the ideas and proof techniques.**
> Our main upper bound algorithm uses both importance sampling and a hierarchical space decomposition. While these tools are indeed quite common, we would like to point out that we apply them in a subtle but crucially different manner than prior works. We hope the following elaboration will be insightful in pointing out the differences.
>
> The value of the Chamfer distance is exactly equal to the sum of distances from each point to its nearest neighbor. We approximate each distance to nearest neighbor using a randomized hierarchical decomposition; for the case of the L1 norm, each level of the decomposition partitions the space into rectangular boxes, as in quadtrees.  Crucially, however, to ensure that the running time of our algorithm is within the stated bounds, we cannot use the standard randomly shifted quadtree where each level is shifted by the same random vector, a common method in geometric approximation algorithms. This is because shifting all levels by the same amount only ensures that the expected distortion between a fixed pair of points is logarithmic; to ensure that the distance to the nearest neighbor is distorted by O(log n), we would need to use O(log n) independent quadtrees and apply the union bound, leading to extraneous overhead. Instead, we use independent random partitions at each level, and show (Lemma 2.3) that this suffices to bound the expected distortion of the distance to the nearest neighbor, without incurring any additional factors. This makes it possible to obtain the running time as stated.
>
> We will add the comparison to the final version of the paper.

---

> > ### Comment · Reviewer_3E3v · 2023-08-17
> > **Replying to Rebuttal**
> >
> > Thanks for the response. For now I maintain my score

---

### Official Review · Reviewer_cYq9 · 2023-07-25

**Soundness:** 3 good
**Presentation:** 3 good
**Contribution:** 3 good
**Rating:** 7
**Confidence:** 3

**Summary:**

This paper studies the approximation algorithm for computing the Chamfer distance between two datasets. The Chamfer distance is a very useful distance metric in machine learning, and the naive approach for computing it takes $O(n^2d)$-time for $n$ points in $d$ dimensions. This paper proposes an approximation algorithm that runs in $O(nd\log(n)/\epsilon^2)$-time with $(1+\epsilon)$-relative error. This algorithm is the first to achieve linear dependence in $nd$ and polynomial dependence in $1/\epsilon$. Moreover, it is implementable on real datasets and empirically improves the previous methods. In addition, it shows a fine-grained lower bound that no randomized algorithm can find a $(1+ \epsilon)$-approximate mapping for the Chamfer distance in $n^{2-o(1)}$-time for $d\sim \log^2 n$ and $\epsilon\sim 1/d$, assuming the Hitting Set Conjecture. Technically, their algorithm is based on the importance sampling and Locality-Sensitive Hashing.


**Strengths:**

The problem studied in this paper is significant in theory and practice. The algorithm in this paper runs in nearly-linear time in $n$ and works for large dimensions and high accuracy. For comparison, previous approximation algorithms either incur an exponential dependence in $d$ or only work for the low-accuracy regime. This paper provides an essential theoretical improvement. Another advantage is that the algorithm is quite clean that can be implemented efficiently, which increases the practical value of this paper.  This paper thoroughly explains the experiments.  And most theorems and lemmas in this paper are mathematically sound to me.


**Weaknesses:**

Some notations could be more precise in the main text. For example, the subroutine CrudeNN uses $H(r)$ without defining it. And more intuition should be given for this subroutine. Also, the algorithm in this paper works for any distance metric with LSH functions. Some examples other than $\ell_1$ or $\ell_2$ can be provided. For the hardness result, the Hitting Set Conjecture is not very standard in the field of fine-grained complexity.


**Questions:**

1. In Section 2, it discusses the uniform sampling approach. And the runtime is $O(nds)$ if we sample $s$ points. What if we use any computational geometry data structure to pre-process the set $B$ such that for each sample, the computational cost can be reduced to $o(nd)$?

2. Line 27: $O(n^2)$ should be $O(n^2d)$ to avoid confusion.

3. Line 85: ``distribution, $D_a$”. However, $D_a$ itself is not a probability distribution but an estimator.

4. Proof of Lemma A.2: the event $E_1(\gamma)$ is missing.

---

> ### Author Rebuttal · Authors · 2023-08-08
>
> **On notation.** We apologize for the lack of precision in notation in the main text. We will fix these issues.
>
> **On metrics that admit a locality-sensitive hash.** We focused on L2/L1 as, to our knowledge, these are the most typical cases. However, our methods work for any metrics which support a locality sensitive hash function (or such metrics squared). This encompasses a very wide class of metrics including $\ell_1, \ell_2$ (and generally all $\ell_p$ for $1\le p \le 2$), the Jaccard distance, and even includes functions which are not strictly metrics such as $\ell_p$ for $0 < p < 1$ and $\ell_2^2$.  We will make it clear in the updated version of the paper.
>
> **The Hitting Set Conjecture.** Although the conjecture is not as popular as, say, Strong Exponential Time Hypothesis, it has nevertheless been used in many papers, as listed below. Thus, we believe that hardness results assuming HSC provide useful evidence of problem complexity.
>
> [1] Approximation and fixed parameter subquadratic algorithms for radius and diameter in sparse graphs, Abboud et al,  SODA 2016.
>
> [2] Approximation algorithms for min-distance problems, Dalirrooyfard et al, ICALP 2019.
>
> [3] Tight approximation algorithms for bichromatic graph diameter and related problems, Dalirrooyfard et al ICALP 2019.
>
> [4] Approximation algorithms for min-distance problems in DAGs, Dalirrooyfard and Kaufmann, ICALP 2021.
>
> [5] Conditional hardness of earth mover distance, Rohatgi, APPROX 2019.
>
> [6] Completeness for first-order properties on sparse structures with algorithmic applications, Gao et al, SODA 2017.
>
> [7] Dynamic Data-Race Detection Through the Fine-Grained Lens, Kulkarni et al, CONCUR 2021.
>
> [8] Fast approximation of eccentricities and distances in hyperbolic graphs, Chepoi et al, Journal of Graph Algorithms and Applications, 2019.
>
> [9] Tight hardness results for distance and centrality problems in constant degree graphs, Dahlgaard and Evald, arxiv preprint, 2016.
>
> [10] On the complexity of 1 center in various metrics, Abboud et al, arXiv preprint, 2021.
>
> [11] Balancing graph Voronoi diagrams with one more vertex, Ducoffe, 2022
>
>
> **What if we use any computational geometry data structure to pre-process the set such that the computational cost can be reduced to o(nds)?**
>
> We assume that the reviewer suggests using a nearest neighbor data structure that computes an (approximate) nearest neighbor of each uniformly sampled point. Indeed, we discuss a very similar idea on page 2, lines 30-34, though the data structure is applied to *all* n points. Unfortunately, unless $s=\Omega(n)$, we do not believe uniform sampling will yield provable guarantees with good probability. This is because the distance from one of the points (say p) in A to its nearest neighbor could be very large, and if the sample misses p, the estimate will be inaccurate.
>
> **Line 27: $O(n^2)$ should be $O(n^2d)$ to avoid confusion. Line 85: ``distribution $D_a$”. However $D_a$ itself is not a probability distribution but an estimator.**
>
> We thank the reviewer for pointing out these typos, we will fix them.
>
> **Proof of Lemma A.2: the event $E_1(\gamma)$ is missing.**
>
> We apologize for the typo, the definition of the event was omitted while preparing the supplementary. The event $E_1(\gamma)$ occurs when there exists a point $b' \in B$ at distance at least $\gamma$ from $a$ and there exists an index $i \leq i_0$ for which $h_i(a) = h_i(b')$.

---

> > ### Comment · Reviewer_cYq9 · 2023-08-14
> >
> > Thank you for your response. I’ll keep my score.

---

### Official Review · Reviewer_WXaW · 2023-07-26

**Soundness:** 4 excellent
**Presentation:** 4 excellent
**Contribution:** 3 good
**Rating:** 7
**Confidence:** 5

**Summary:**

This paper considers the problem of computing the Chamfer distance between two point sets. Informally, the Chamfer distance is simply the total cost of mapping points of A to their closest point in B. Practically, Chamfer distance seems to be used as a proxy for the earth mover’s distance.

One can always compute Chamfer distance is quadratic time – for every point in A, find the cheapest edge by simply scanning all the edges. Of course, in low dimensional spaces, one can speed this up by the use of exact nearest neighbor data structures. To approximate the Chamfer distance, one can use the approximate nearest neighbor data structure leading to a near-linear time algorithm in fixed dimension and a super-linear time algorithm in higher dimensions.

The main contribution of this paper is to show that Chamfer distance can be estimated in near-linear time even in higher dimensions. As opposed to this, the authors show that computing the maps themselves is hard to approximate in near-linear time (the running time has to be only slightly sub-quadratic under the hitting set conjecture).

The main technique used is the idea of importance sampling. Their algorithm first obtains a crude approximation of the contribution of each point and then uses these values to obtain a small sized weighted sample which can then be used to estimate the Chamfer distance in $O(nd\log n/\varepsilon^2)$ time. The idea of using Importance sampling to estimate distances efficiently is not new. It has been used, for instance, to estimate EMD in low dimensions by Indyk in his paper titled “A near linear time constant factor approximation for Euclidean bichromatic matching (cost).”
Nonetheless, the use of importance sampling for approximate Chamfer distance is novel and the contribution towards computing high dimensional Chamfer distance is significant. For this reason, I recommend accepting this paper.


**Strengths:**

The algorithm is simple and estimates Chamfer distance for high dimensional data with solid theoretical guarantees and strong practical results.

**Weaknesses:**

I’m not convinced that Chamfer distance has strong theoretical properties. For instance, it doesn’t appear to be symmetric, i.e., Chamfer(A,B) is not equal to Chamfer(B,A). Is there a relationship between Chamfer distance and EMD even under some stochastic assumptions for B and A?

Similar techniques seem to have been used for EMD approximations before (Indyk, SODA 2007). Could you add a comparison of your algorithm to the techniques used in other papers?


**Questions:**

Please answer the questions in weaknesses.

---

> ### Author Rebuttal · Authors · 2023-08-07
>
> We thank the reviewer for their detailed and thorough comments. We would first like to remark that, while it is true that Chamfer is usually used as a proxy for EMD, there are scenarios where Chamfer distance is actually the correct choice instead of EMD. This includes settings where duplication of points should not affect the similarity significantly: an example is when the point clouds are sets of pre-trained image embeddings of a location (building, city, ect.) and one wants to determine if the location described by the point cloud  is the same.
>
> **Relationship between Chamfer and EMD:**
> With regards to theoretical properties of the Chamfer distance, note that for symmetry, a common practice is to compute Chamfer(A,B) + Chamfer(B,A), which is symmetric. Our method also easily extends to the symmetric case by two calls to our algorithm. With regards to comparing Chamfer to EMD, it’s always true that EMD(A,B) >= Chamfer(A,B). This fact is particularly useful for finding points in a dataset with EMD smaller than some threshold to a query (nearest neighbor search): first quickly compute the Chamfer distance from each point to the query, and if the Chamfer distance to a point is too large then one does not need to compute the EMD to that point. This is called prefetch and prune (see e.g. “From Word Embeddings To Document Distances” by Kusner, Sun, Kolkin, and Weinberger, ICML '2015).
>
> Note that under stochastic assumptions, the Chamfer distance may be quite close to EMD depending  on the assumptions made. For example, if the point sets A and B are uniform on the unit sphere in at least $\Omega(\log n)$ dimensions, then with high probability, both values are within $1+o(1)$ multiplicative factors of each other since all the points are pairwise distance $\sqrt(2) - o(1)$ apart, with high probability. Further exploring the relationship between Chamfer and EMD for stochastic inputs is an interesting future direction.
>
> **Comparison between our algorithm and other papers:**
> With regards to the comparison of our algorithm to techniques used in prior papers: we note that importance sampling is indeed a popular technique used for speeding up algorithms. In particular, as the reviewer points out, the paper of Indyk, SODA, 2007, uses it to obtain a fast c-approximate algorithm for computing EMD in two (or any constant) dimensions, for some constant c>2. However, the application and implementation of importance sampling in that paper is quite different from ours.
>
> In (Indyk’07), the space containing all input points is subdivided into regions, and the total EMD value is represented as a sum of EMDs restricted to point-sets in each region (plus an additional representing the “global” EMD). The EMD cost in each region is then approximated quickly by embedding EMD into L1 using a randomly shifted quadtree with logarithmic distortion; these estimations define the sampling probabilities.
>
> In contrast, in our paper, the value of the Chamfer distance is exactly equal to the sum of distances from each point to its nearest neighbor, so there is no decomposition involved. Instead, we approximate each distance to nearest neighbor using a randomized hierarchical decomposition; for the case of the L1 norm, each level of the decomposition partitions the space into rectangular boxes, as in quadtrees.  Crucially, however, to ensure that the running time of our algorithm is within the stated bounds, we cannot use a standard randomly shifted quadtree where each level is shifted by the same random vector (as in Indyk’07). This is because shifting all levels by the same amount only ensures that the expected distortion between a fixed pair of points is logarithmic; to ensure that the distance to the nearest neighbor is distorted by O(log n), we would need to use O(log n) independent quadtrees and apply the union bound. Instead, we use independent random partitions at each level, and show (Lemma 2.3) that this suffices to bound the expected distortion of the distance to the nearest neighbor, without incurring any additional factors. This makes it possible to obtain the running time as stated.
>
> We will add the comparison to the final version of the paper.

---

> > ### Comment · Reviewer_WXaW · 2023-08-12
> >
> > Thank you for the response.
> >
> > One quick follow-up. My understanding is Chamfer distance is a lower bound on EMD only if every point of the distribution has a mass of 1/n. Is there a generalization of the definition of Chamfer distance for distributions where support points can have arbitrary real-valued masses? If so, does your algorithm extend to this definition?

---

> > > ### Author Response · Authors · 2023-08-13
> > > **Follow up to reviewer comment**
> > >
> > > Thank you for the response. As per in the references [6,14], one  can generalize the Chamfer distance to the weighted sum $\sum_{a \in A} w_a \cdot \min_{b \in B} d(a, b)$ for non-negative weights $w_a$ for $a \in A$, and this lower-bounds the corresponding weighted version of EMD. Our algorithm also extends to this case since we can multiply our approximation $D_a$ for $\min_{b \in B} d(a, b)$ by $w_a$, which retains the guarantee that it is a $O(\log n)$ multiplicative approximation (in expectation), to the true weighted distance $w_a \cdot \min_{b \in B} d(a, b)$. This is sufficient for importance sampling and implies the same asymptotic runtime as in the uniform weight case.

---

### Author Rebuttal · Authors · 2023-08-08

We thank all the reviewers for useful comments and feedback. We will fix the typos and presentation issues in the final version of the paper. In what follows in the individual rebuttals, we address the issues identified by the reviewers as weaknesses and/or listed as questions.

---

### Decision · Program_Chairs · 2023-09-21

**Decision:**

Accept (poster)

**Comment:**

The paper addresses the computation of the Chamfer distance between two point sets, defined as the cost of mapping points from set A to their nearest points in set B, often used as a proxy for the earth mover's distance. While traditional methods involve quadratic time complexity, this work introduces an algorithm that estimates the Chamfer distance in near-linear time even in higher dimensions using importance sampling. This approach obtains an approximate contribution value for each point, then derives a weighted sample to efficiently estimate the Chamfer distance. Unlike mapping computations, which remain hard to approximate, this method proves valuable for efficiently estimating high-dimensional Chamfer distances, making a notable contribution to the field. Referees were mostly positive and recommend acceptance.